

# Novice assessors demonstrate good intra-rater agreement and reliability when determining pressure pain thresholds; a cross-sectional study

Roland R. Reezigt[1,2], Geranda E. C. Slager[2], Michel W. Coppieters[1,3,4] and Gwendolyne G. M. Scholten-Peeters[1]

[1] Department of Human Movement Sciences, Faculty of Behavioural and Movement Sciences, Amsterdam Movement Sciences—Program Musculoskeletal Health, Vrije Universiteit Amsterdam, Amsterdam, Netherlands
[2] Academy of Health, Department of Physiotherapy, Hanze University of Applied Scienses, Groningen, Netherlands
[3] Griffith University, Menzies Health Institute Queensland, Brisbane and Gold Coast, Australia
[4] School of Health Sciences and Social Work, Griffith University, Brisbane and Gold Coast, Australia

Corresponding author
Gwendolyne G. M. Scholten-Peeters, g.g.m.scholten-peeters@vu.nl

## ABSTRACT

**Background:** Experienced assessors show good intra-rater reproducibility (within-session and between-session agreement and reliability) when using an algometer to determine pressure pain thresholds (PPT). However, it is unknown whether novice assessors perform equally well. This study aimed to determine within and between-session agreement and reliability of PPT measurements performed by novice assessors and explored whether these parameters differed per assessor and algometer type.

**Methods:** Ten novice assessors measured PPTs over four test locations (tibialis anterior muscle, rectus femoris muscle, extensor carpi radialis brevis muscle and paraspinal muscles C5-C6) in 178 healthy participants, using either a Somedic Type II digital algometer (10 raters; 88 participants) or a Wagner Force Ten FDX 25 digital algometer (nine raters; 90 participants). Prior to the experiment, the novice assessors practiced PPTs for 3 h per algometer. Each assessor measured a different subsample of ~9 participants. For both the individual assessor and for all assessors combined (*i.e.*, the group representing novice assessors), the standard error of measurement (SEM) and coefficient of variation (CV) were calculated to reflect within and between-session agreement. Reliability was assessed using intraclass correlation coefficients ($ICC_{1,1}$).

**Results:** Within-session agreement expressed as SEM ranged from 42 to 74 kPa, depending on the test location and device. Between-session agreement, expressed as SEM, ranged from 36 to 76 kPa and the CV ranged from 9–16% per body location. Individual assessors differed from the mean group results, ranging from −55 to +32 kPa or from −9.5 to +6.6 percentage points. Reliability was good to excellent ($ICC_{1,1}$: 0.87 to 0.95). Results were similar for both types of algometers.

**Conclusions:** Following 3 h of algometer practice, there were slight differences between assessors, but reproducibility in determining PPTs was overall good.

## INTRODUCTION

Since Victorian times, attempts have been made to quantify pain perception with mechanical stimulus devices (*Keele, 1954*). The measurements and devices evolved into analogue and digital pressure algometers to obtain pressure pain thresholds (PPTs) (*Nussbaum & Downes, 1998*). PPT is defined as the minimal applied pressure that induces a painful sensation (*Fischer, 1987*). PPTs are often used as a static measure of pain sensitivity in various patient populations in research (*Amiri et al., 2021*) and clinical practice (*Ohrbach & Gale, 1989*; *Nijs et al., 2021*). More recently, PPTs also became part of dynamic measures, such as conditioned pain modulation (CPM). In CPM, a baseline PPT is compared to a PPT during or after a conditioned stimulus (*Yarnitsky et al., 2015*; *Reezigt et al., 2021*).

A standardised protocol to determine PPTs is available within a Quantitative Sensory Testing (QST) framework to optimise reproducibility (*Rolke et al., 2006*; *Mücke et al., 2016*). Reproducibility relates to the degree to which repeated measurements in stable individuals provide similar results (*de Vet et al., 2006*). Intra-rater reproducibility, where test results of the same rater are analysed, can be further defined into an absolute test-retest parameter called agreement (or measurement error), and a relative test-retest parameter called reliability (*Stratford & Goldsmith, 1997*; *Bruton, Conway & Holgate, 2000*; *Mokkink et al., 2010*). Agreement can be further specified into within-session agreement and between-session agreement. Each PPT is determined as the average of multiple repeated ratings (typically two to four measurements) within one session. Within-session agreement indicates how equal these ratings within the measurement session are, reflecting the precision of the measurement. Between-session agreement represents how close each averaged score for repeated measurements is (*i.e.*, the measurement error), which is important for evaluation purposes. Reliability, on the other hand, accounts for between-subject variability within the tested sample. It shows how well the test can distinguish people from each other, which is essential for diagnostic purposes (*de Vet et al., 2006*).

Generally, PPTs measured with a digital algometer show good to excellent intra-rater agreement and reliability for most body locations when an experienced rater performs the measurements. For example, for agreement, a standard error of measurement (SEM) of 63 to 88 kPa on larger muscles associated with a higher PPTs, such as the tibialis anterior muscle, and 41 to 53 kPa on smaller muscles wither lower PPTs, such as the paraspinal muscles in the neck region, and good coefficients of variations (CVs) (*e.g.*, 8% to 19% for tibialis anterior muscle, and 14% to 18% for the neck region) have been reported. Furthermore, the reported reliability is high (*e.g.*, intraclass correlation coefficients (ICCs) of 0.79 to 0.94 for the tibialis anterior muscle (*Sterling et al., 2002*; *Jørgensen et al., 2014*;

*Srimurugan Pratheep, Madeleine & Arendt-Nielsen, 2018*), and 0.82 to 0.91 for the neck region (*Sterling et al., 2002*; *Ylinen et al., 2007*; *Knapstad et al., 2018*)).

Agreement and reliability are generally reported as moderate to good in both larger and smaller muscles. However, the reported parameters vary between studies. This variability can partly be attributed to rater characteristics as PPT measurements seem to be operator dependent. Determining PPTs requires practical and psychomotor skills of the individual rater. For example, variability in application rate of the pressure increase negatively impacts on reliability and agreement (*Linde et al., 2017*). Acquiring competence in such skills can be achieved through guided practice, repetition and reinforcement, going through the cognitive, associative and autonomous phases of motor learning (*Schmidt & Lee, 2005*; *Tynjälä & Gijbels, 2012*; *Oermann, Muckler & Morgan, 2016*; *Reilly, Beran-Shepler & Paschal, 2020*). Novice raters may still be developing their psychomotor skills, so it could be assumed that they have not yet achieved comparable skills to experienced raters (*Sattelmayer et al., 2016*). In studies examining reliability of PPTs, only one study included novice raters (*Walton et al., 2011*). This study showed lower reliability values (ICC 0.79) compared to studies with experienced raters (ICC 0.92) (*Waller et al., 2016*). As individual psychomotor skills differ between people (*Schmidt & Lee, 2005*; *van Duijn, Swanick & Donald, 2014*), this may lead to different reproducibility parameters per individual. As such, it remains unclear whether novice raters are able to determine PPT measurements with an acceptable reproducibility, following limited hours of training. Insight into reproducibility is important as QST measures are often performed by novice raters in student projects and higher degree research projects (*Geber et al., 2009*). Simply extrapolating reproducibility parameters from experienced raters is debatable.

To summarise, research about agreement and reliability parameters in novice raters when performing PPTs is scarce despite the fact that novice raters often perform these measurements. Therefore, this study aimed to determine intra-rater reproducibility parameters (within-session and between-session agreement and reliability) of PPTs measured by novice (*i.e.*, inexperienced) raters. Additionally, this study explored the extent of individual differences in intra-rater reproducibility parameters.

## METHODS

### Design

Using a cross-sectional, observational test-retest study design, ten novice raters determined PPTs using two types of digital algometers to assess intra-rater reproducibility parameters. For agreement, the SEM was used as absolute value for the measurement error and the CV was used as relative value for the measurement error compared to the mean PPT of the muscle (*Mokkink et al., 2010*). For reliability, the $ICC_{1,1}$ was used as ratio between the measurement error (*Mokkink et al., 2010*). All intra-rater reproducibility parameters were measured per algometer.

Data were acquired between February and June 2019. The study was approved by the Medical Ethical Committee of the University Medical Center in Groningen, The Netherlands (METc 2016.613; M17.207169). All participants and raters signed an informed consent prior to the measurements. Reported reproducibility parameters are

based upon the COSMIN group recommendations for studies on psychometrics (*Gagnier et al., 2021*; *Mokkink et al., 2010*; *de Vet et al., 2016*). The Guideline for Reporting Reliability and Agreement Studies (GRRAS) was used (*Kottner et al., 2011*; *Gerke et al., 2018*).

## Raters

Novice raters were students in the last year of their undergraduate physiotherapy program. Ten raters (six females, four males) were recruited from the Hanze University of Applied Sciences in Groningen, The Netherlands. Physiotherapy students without experience in determining PPTs were eligible to participate. Exclusion criteria were being physically unable to perform PPTs on the selected muscles (in terms of positioning and strength) or not being able to communicate in the Dutch language. Their mean (SD) age, weight and height were 23.0 (3.0) years, 75 (11) kg and 179 (6) cm. For this project, all raters attended two four-hour training session to perform PPT measurements with a Somedic and a Wagner digital algometer. The training included an explanation of the study procedures, standardised participant instructions, palpating and marking of the body locations (approximately 2 h) and the use of the two digital algometers (for approximately 3 h per algometer). Practising assessment of PPTs was performed on healthy participants who did not participate in the main study.

## Participants

Pain-free, healthy participants were recruited from the staff and student populations of the Hanze University of Applied Sciences, Academy of Health Care, in Groningen, The Netherlands, and the local community. The selection of healthy participants was based on the aim to determine rater-dependence in PPT measurements and as such to eliminate potential confounding variables of dysfunction or disease. Convenience sampling took place *via* an announcement on the university's intranet and an email was sent to all staff and students. All participants underwent a short screening to determine eligibility (*Gierthmühlen et al., 2015*). Inclusion criteria were being pain-free, between 18 and 65 years old and an adequate understanding of the Dutch language. Participants were not permitted to use alcohol, caffeine or nicotine-containing products 2 h before testing, or using analgesics 24 h prior to testing (*Girdler et al., 2005*; *Baratloo et al., 2016*; *Bagot et al., 2017*). Exclusion criteria were the presence of acute or chronic pain, neurological, orthopedic or cognitive disorders, pregnancy or if participants had undergone surgery to the legs, arms or neck.

## Sample size

Sample size estimation was performed for both agreement and reliability parameters for the group of raters. For agreement, using an upper bound of the 95% CI of +15% of the SEM, a minimum of 81 participants was required (per algometer) (*Stratford & Goldsmith, 1997*). For reliability, based on $\alpha = 0.05$, $\beta = 0.8$, an expected $ICC_{1,1}$ of >0.7 with the lower bound of the 95% CI of 0.5, a minimum of 63 participants was required to calculate reliability per algometer (*Walter, Eliasziw & Donner, 1998*).

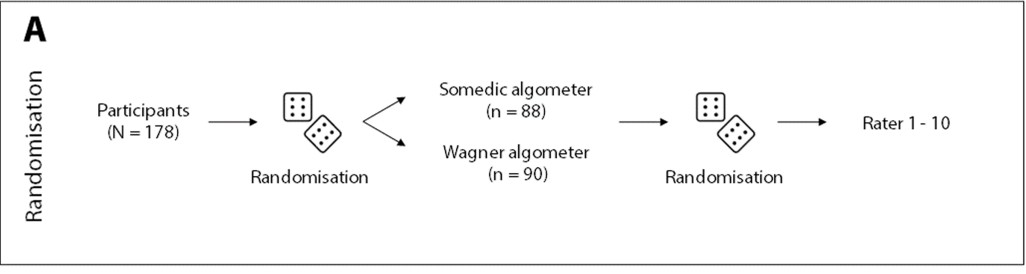

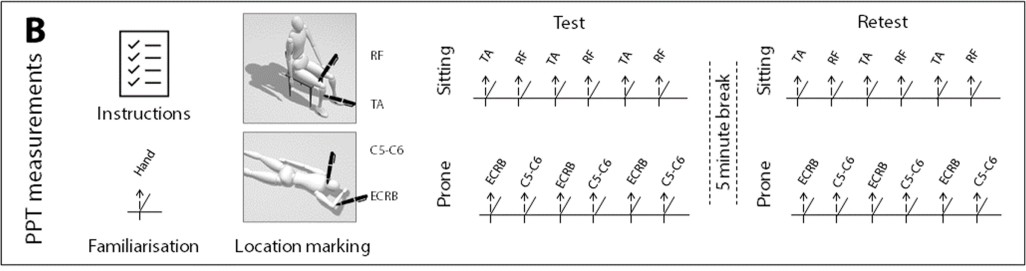

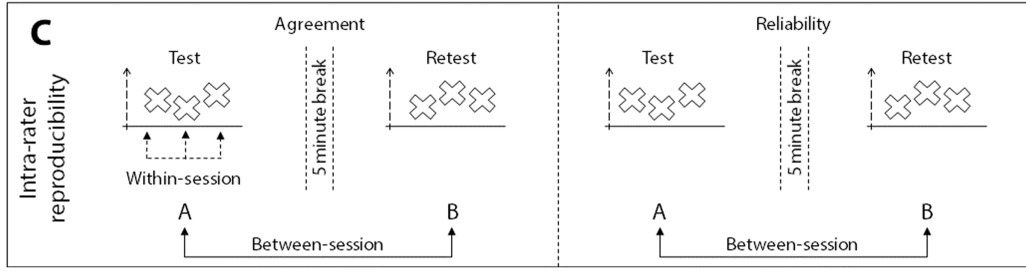

**Figure 1 Visual representation of the methods.** (A) The randomisation procedure for algometer type and rater allocation; (B) shows the measurement procedure; (C) illustrates which measurements are used for the within-session and the between-session calculations. The mean of the three measurements were used for the between-session calculations. RF, rectus femoris muscle; TA, tibialis anterior muscle; C5-C6, paraspinal muscles at C5-C6; ECRB, extensor carpi radialis brevis muscle.

## PPT measurements

After collecting the demographic, anthropometric, lifestyle and psychosocial data, participants were randomly allocated to a rater and went into one of four rooms where they received standardised instructions. Then, every participant was familiarised with the measurement on the dorsal aspect of the hand by the allocated rater. Figure 1 shows the procedure and methods used in the study. Four rooms were used to simultaneously measure all participants. The rooms were comparable regarding size, temperature and noise levels.

PPTs were measured using either a Somedic digital algometer (Type II; Somedic AB, Stockholm, Sweden) or Wagner digital algometer (Force Ten FDX 25; Wagner, Greenwich, CT, USA). Both digital algometers were used with a 1 cm$^2$ rubber tip with an application rate of ~50 kPa/s. The Somedic algometer provides visual feedback regarding the application rate and has a hand-held switch for the participant to press when the feeling of pressure changes into painful pressure. The Wagner device does not provide feedback

regarding the application rate. Consequently, the raters had to estimate the rate themselves. Because visual feedback about the slope might assist raters in obtaining more stable PPTs, the two different algometers were used to assess whether the intra-rater reproducibility parameters are influenced by the algometer properties. Furthermore, the Wagner algometer has no patient switch, so participants verbally indicated when their pain threshold was reached.

After marking the locations with a skin pencil, raters measured the PPTs three times in each location, with an inter-stimulus interval of 20 s to avoid sensitisation of pain (*Reezigt et al., 2021*; *Nussbaum & Downes, 1998*). The measurements were taken on the dominant side, in a fixed order in two circuits. In the first circuit, with the participant in sitting, PPTs were determined on the tibialis anterior muscle (~5 cm distal to the tibial tuberosity) and rectus femoris muscle (~15 cm proximal to the patella base) on the dominant side, alternating between the two locations (Fig. 1B). In the second circuit, with the participant in prone and the dominant arm above the head, PPTs were determined between extensor carpi radialis brevis muscle (~5 cm distal to the lateral epicondyle of the humerus) and the paraspinal muscles at the C5-C6 level of the neck, again in an alternating manner (Fig. 1B). The test locations were selected based on the different levels of complexity of the measurements and height of the PPT. Furthermore, these muscles are often selected in PPT studies and their reproducibility parameters are available for experienced raters (*Sterling et al., 2002*; *Ylinen et al., 2007*; *Jones, Kilgour & Comtois, 2007*; *Jørgensen et al., 2014*; *Waller et al., 2016*; *Jakorinne, Haanpää & Arokoski, 2018*; *Srimurugan Pratheep, Madeleine & Arendt-Nielsen, 2018*; *Knapstad et al., 2018*; *Middlebrook et al., 2020*).

After removal of the skin markings and a 5-min wash-out period, the rater re-marked the locations and retested the PPTs.

## Randomisation and Blinding

Participants were randomly allocated to either being measured (1) with the Somedic or Wagner algometer, and (2) by one of the ten raters. Computer-generated randomisation and opaque envelopes were used. To blind the raters for the measurement results, they handed the algometer to a research assistant after each measurement. The research assistant read the value on the display, recorded the result, reset the algometer and handed the device back to the rater. Participants were also blinded for their test values by shielding the display of the algometer during the measurements.

## Statistical analysis

### Descriptive statistics

Descriptive analyses were used to report participant characteristics containing demographic, anthropometric, lifestyle and psychosocial data. The normality assumption was visually inspected and tested using the Shapiro–Wilk test. Normally distributed data were described as mean (SD). All statistical analyses were performed in SPSS version 28 (IBM, Armonk, NY, USA). Participant's characteristics measured in the Somedic algometer group and Wagner algometer group were analysed to investigate comparability of the two samples and to be able to indirectly compare the reproducibility parameters

between the two algometers. Continuous data were tested using either an independent t-test (in normal distribution) or non-parametric Mann–Whitney U Test. The Chi-Square Test was used for categorical data. All PPT data of the Somedic algometer were expressed in kPa. All PPT data of the Wagner algometer were converted from Newton/cm$^2$ to kPa.

### Reproducibility parameters

All reproducibility parameters were intra-rater parameters and calculated per algometer. Group reproducibility parameters reflect novice raters as a group, whereas individual reproducibility parameters reflect each novice rater individually. Intra-rater absolute (*i.e.*, SEM) and relative (*i.e.*, CV) parameters were calculated for both within-session and between-session agreement. The $ICC_{1,1}$ was calculated for reliability. For all analyses, per algometer and per location, outlier or influential case detection and removal were based on average PPT and difference scores greater than three SDs (reducing an overrepresentation of extreme cases in a normal distribution) (*Parrinello et al., 2016*).

### Within-session agreement

Within-session agreement reflects the precision within a measurement session; *i.e.*, how close the three ratings of the first test session are (Fig. 1C). First, for both the group and each individual rater, the within-session agreement was calculated as an absolute value in the form of the SEM. To eliminate the bias of subject variability, the calculation was based on the error mean square term (EMS) of the Analysis of Variance (ANOVA) used for the $ICC_{1,1}$ calculations, according to the formula: $SEM = \sqrt{EMS}$ (*Weir, 2005*). Confidence intervals were calculated based on the sample error variance; $\left[\sqrt{\frac{SSE}{\chi^2_{\alpha,dfe}}}; \sqrt{\frac{SSE}{\chi^2_{1-\alpha,dfe}}}\right]$, where SSE is the sum of squares error (from the ANOVA), $\chi^2_{\alpha,dfe}$ is the chi-square value for significance level $\alpha$ (0.05) and the corresponding degrees of freedom (df) of the SSE term (*Stratford & Goldsmith, 1997*; *Armitage & Berry, 2001*).

Second, as each sample of ratings (per location, per rater) may differ from the average PPT (*e.g.*, PPT ratings on the leg are higher than those on the arm), the CV, also called the relative measurement error (*i.e.*, SEM%), is used as an accompanying agreement parameter to increase comparability between locations and raters. The CV is calculated as the SEM divided by the sample average threshold and is reported as a percentage:

$$Within - session\ CV = \frac{SEM_{Within-session}}{AVG(3\ ratings)} * 100 \qquad , where$$

$$SEM_{Within-session} = \sqrt{EMS_{Within-session}}$$

A lower SEM and lower CV reflect a better within-session agreement and as such a higher precision (*Atkinson & Nevill, 1998*).

### Between-session agreement

Between-session agreement reflects how close the differences were between the test and retest average ratings (Fig. 1C). For the test and retest sessions, the average of the three measurements was used to calculate the PPT on each location. Subsequently, the

between-session agreement could be calculated as the absolute SEM and relative CV. Additionally, the minimal detectable change of 90% ($MDC_{90}$) based on the SEM (*Weir, 2005*) was calculated for clinical purposes (*Donoghue et al., 2009*) by:

$$MDC_{90} = SEM * \sqrt{2} * 1.65$$

### Dose-dependent differences

When dose-dependent differences are present, a relative parameter (*e.g.*, CV) reflects the measurement error better than an absolute parameter (*e.g.*, SEM) (*Atkinson & Nevill, 2000*). As such, the data were analysed to see whether the differences between the test and retest increased proportionally to the average threshold increment (*i.e.*, heteroscedasticity of the data; *e.g.*, when more force is needed to determine the PPT, the difference between the test and retest may be larger compared to low force PPT thresholds due to less accuracy when using more strength). First, Bland–Altman plots were created to visualise and analyse the potential dose-dependent differences (*Bland & Altman, 1999*; *Carkeet, 2015*; *Gerke, 2020*).

Second, linear regression lines, based on the absolute value of the differences (*i.e.*, the distance from zero) between the test and retest PPT values, were added to the Bland–Altman plots. The regression analyses were used to estimate the dose-dependent difference more accurately than visual inspection by using the regression coefficient and adjusted $R^2$ (variance explained as an indicator of the strength of the relationship between increasing thresholds and increasing differences).

In this study, presence of dose-dependent differences was defined as: (a) the slope of the regression line was >0.1 and (b) the adjusted explained variance ($R^2$) by the regression model was >0.1. These regression analyses were partly based on suggestions of *Atkinson & Nevill (2000)* and the method of *Ho (2018)*, where we adjusted dose-dependent systematic bias to dose-dependent differences (*Atkinson & Nevill, 2000*; *Ho, 2018*). For these linear regression analyses, per algometer and per location, outlier or influential case detection and removal were based on standardised residuals from the regression greater than three (*Parrinello et al., 2016*).

### Reliability

Reliability reflects the measurement error relative to the subject variability. Reliability estimates were calculated using ANOVA-based ICCs (*Koo & Li, 2016*). The intra-rater reliability represents novice raters in general, and as such each participant was rated by a different rater who was randomly chosen from the set of ten raters. Therefore, ICC estimates and their 95% CI were calculated based on a single rater, absolute agreement, one-way random-effects model ($ICC_{1,1}$) (*Koo & Li, 2016*).

Interpretation of ICCs is based on the criteria of *Koo & Li (2016)*: values less than 0.5 indicate poor reliability, 0.50 till 0.75 indicate moderate reliability, 0.75 till 0.90 indicate good reliability and values greater than 0.90 indicate excellent reliability.

**Table 1 Characteristics of the participants.**

|  | Somedic algometer group | Wagner algometer group | *P*-value |
|---|---|---|---|
| **Demographics** | *n* = 88 | *n* = 90 |  |
| Sex, male (%) | 44 (50%) | 49 (54%) | $p = 0.65^b$ |
| Age, years | 22 (20–25)[a] | 22 (20–23)[a] | $p = 0.50^c$ |
| **Anthropometrics** |  |  |  |
| Height, cm | 179 (11) | 178 (9) | $p = 0.35$ |
| Weight, kg | 74 (12) | 75 (13) | $p = 0.58$ |
| BMI, kg/m² | 22.9 (2.7) | 23.6 (3.2) | $p = 0.12$ |
| **Lifestyle** |  |  |  |
| Sports, hours/week | 5 (2.5–6.5)[a] | 5 (3.0–7.5)[a] | $p = 0.69^c$ |
| Alcohol use, AUDIT (total score) | 8.0 (4.5) | 8.2 (4.4) | $p = 0.93$ |
| Smoking, *n* (%) | 6 (6.8%) | 16 (17.8%) | $p = 0.04^{b*}$ |
| Sleep, PSQI (total score) | 4.3 (1.9) | 4.7 (2.4) | $p = 0.28$ |
| **Psychosocial** |  |  |  |
| Anxiety, GAD-7 (total score) | 2 (1–3)a | 2 (0–3)[a] | $p = 0.68^c$ |
| Depression, CES-D (total score) | 3 (1–6)[a] | 4 (2–6)[a] | $p = 0.38^c$ |

Note:
Data are presented as mean (SD) unless otherwise specified, [a]Median (IQR), [b]Chi Square Test, [c]Mann–Whitney U Test. *Significant difference between both groups: $\chi^2_{(1, n = 178)} = 4.93$, $p = 0.04$. BMI, Body mass index; AUDIT, Alcohol Use Disorders Identification Test; PSQI, Pittsburgh Sleep Quality Index; GAD-7, Generalised anxiety disorder 7 items; CES-D, Center for Epidemiologic Studies Depression Scale.

## RESULTS

Ten raters measured a total of 178 participants; 88 participants were measured with the Somedic algometer and 90 with the Wagner algometer. Both groups were similar in participant characteristics and baseline PPTs, except for smoking status, which was significantly higher in the Wagner algometer group (Table 1). One rater's data (rater 10) using the Wagner algometer, were excluded as only two participants were measured with this device due to illness of this rater. As a result, the exploration of individual data using the Wagner algometer contains nine raters.

### Group reproducibility

All within-session and between-session agreement and reliability values for the Somedic and Wagner algometer are presented in Table 2. Within-session agreement, as SEM, ranged from 42 to 72 kPa, and as CV from 10.8% to 14.5% for the Somedic algometer. For the Wagner algometer, it ranged from 42 to 74 kPa (SEM), and from 11.5% to 14.8% (CV). Measurements at the rectus femoris muscle showed the best within-session agreement, whereas the measurements at the extensor carpi radialis brevis muscle showed the lowest within-session agreement using either algometer (Fig. 2).

Between-session agreement ranged from 36 to 71 kPa (SEM) and from 10.4% to 16.1% (CV) for the Somedic algometer, and from 47 to 76 kPa (SEM) and from 9.4% to 14.6% (CV) for the Wagner algometer. The best between-session agreement was found at the

**Table 2 Reproducibility parameters on group level.**

| Somedic algometer | Within-session agreement | | | Between-session agreement | | | Between-session reliability |
|---|---|---|---|---|---|---|---|
| | Mean PPT (kPa) | Standard error of measurement (kPa, 95% CI) | Coefficient of variation (95% CI) | Standard error of measurement (kPa, 95% CI) | Minimal detectable change 90% (kPa, 95% CI) | Coefficient of variation (95% CI) | Intraclass correlation coefficient $_{1,1}$ (95% CI) |
| Tibialis anterior | 578 | 61 [55–70] | 11.3% [10.0–12.9%] | 71 [64–82] | 166 [149–191] | 12.9% [11.5–4.8%] | 0.91 [0.87–0.94] |
| Rectus femoris | 600 | 61 [55–70] | 10.8% [9.6–12.4%] | 60 [53–68] | 140 [124–159] | 10.4% [9.2–11.9%] | 0.93 [0.89–0.95] |
| Extensor carpi radialis brevis | 402 | 56 [50–64) | 14.5% [12.9–16.7%] | 63 [56–72] | 147 [131–168] | 16.1% [14.3–18.4%] | 0.89 [0.84–0.93] |
| Paraspinal muscles C5-C6 | 323 | 42 [37–48] | 13.7% [12.1–15.6%] | 36 [32–41] | 84 [75–96] | 11.3% [10.1–13.0%] | 0.93 [0.89–0.95] |
| Wagner algometer | | | | | | | |
| Tibialis anterior | 579 | 74 [66–85] | 13.0% [11.5–14.9%] | 76 [67–87] | 177 [156–203] | 13.1% [11.6–15.0%] | 0.88 [0.83–0.92] |
| Rectus femoris | 655 | 74 [66–85] | 11.5% [10.2–13.2%] | 62 [55–71] | 145 [128–166] | 9.4% [8.4–10.8%] | 0.95 [0.92–0.97] |
| Extensor carpi radialis brevis | 364 | 52 [46–60] | 14.8% [13.2–17.0%] | 47 [42–54] | 110 [98–126] | 12.9% [11.5–14.8%] | 0.90 [0.85–0.93] |
| Paraspinal muscles C5-C6 | 325 | 42 [38–49] | 13.4% [11.9–15.4%] | 47 [42–54] | 110 [98–126] | 14.6% [13.0–16.7%] | 0.87 [0.81–0.92] |

**Note:**
kPa, kilo Pascal.

rectus femoris muscle using either algometer (Fig. 2). Measurements at the extensor carpi radialis brevis muscle showed the lowest agreement for the Somedic algometer (CV: 16.1% (95% CI [14.3–18.4%]) on an average PPT of 393 kPa), whereas measurements at the paraspinal muscles at C5-C6 were the lowest for the Wagner algometer (CV: 14.6% (95% CI [13.0–16.7%]) on an average PPT of 325 kPa). The values of the $MDC_{90}$ are presented in Table 2.

Dose-dependent differences (higher average thresholds associated with larger differences) were found at the tibialis anterior muscle (slope of 0.12x, $R^2 = 0.301$) and the extensor carpi radialis brevis muscle (slope of 0.15x, $R^2 = 0.253$) using the Somedic algometer. Using the Wagner algometer, dose-dependent differences were present at the paraspinal muscles of C5-C6 (slope of 0.17x, $R^2 = 0.225$). Measurements at the rectus femoris muscle showed the lowest dose-dependent differences (Figs. 3 and 4).

The reliability was good to excellent at all locations with both algometers. The $ICCs_{1,1}$ for the Somedic algometer ranged from 0.89 to 0.93, and for the Wagner algometer from 0.87 to 0.95 (Fig. 2).

## Exploration of individuals' reproducibility

Individual within-session agreement expressed as SEM ranged from 24 to 120 kPa, depending on the rater, location and algometer. Consequently, the maximal individual offset compared to the group findings was −26 and +46 kPa (both on the rectus femoris muscle). The CV ranged from 5.1% to 21.3%, with maximal differences compared to the group findings of −6.0 to +9.3 percentage points (both on the rectus femoris muscle).

Individual between-session agreement expressed as SEM, ranged from 15 to 115 kPa, depending on the rater, location and algometer. Compared to the group findings, the

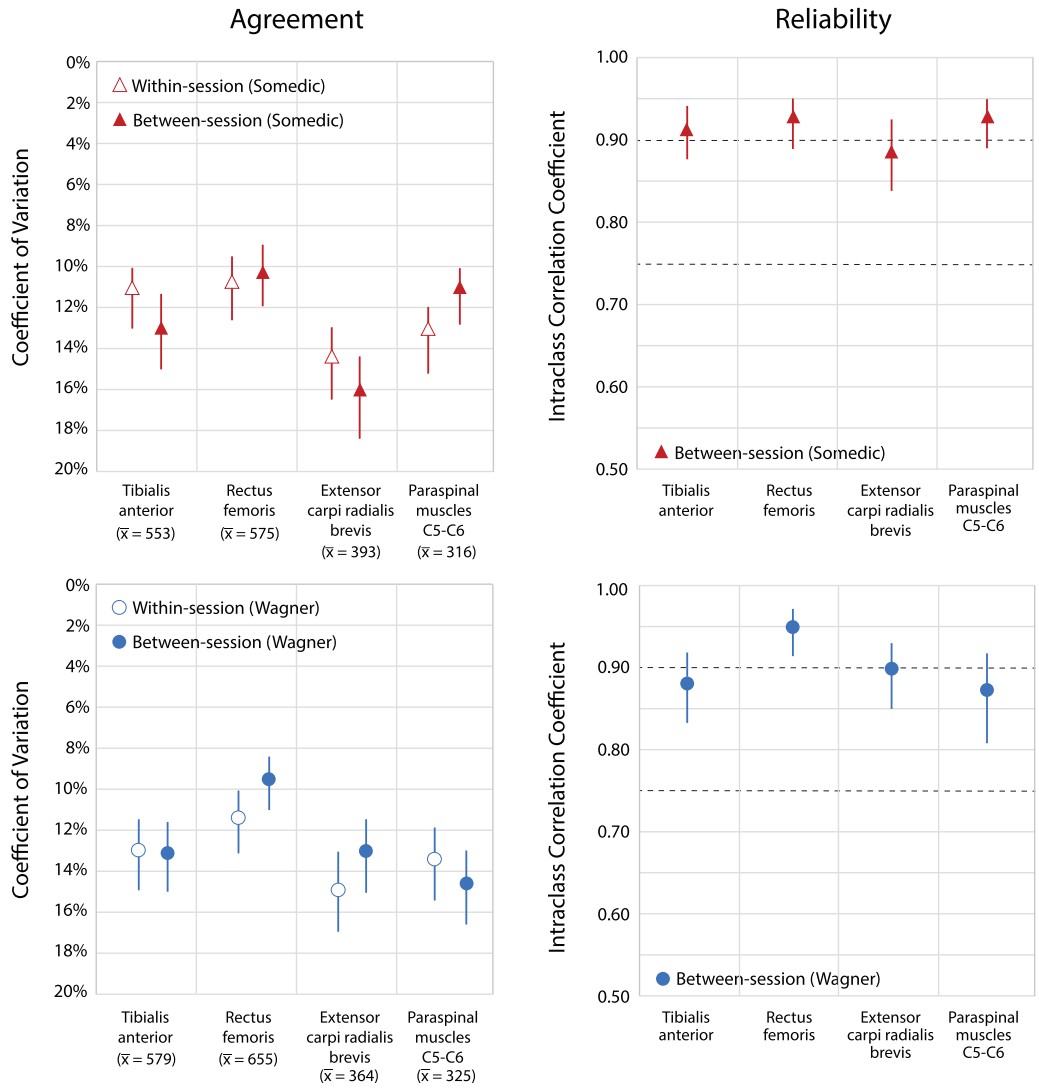

**Figure 2 Within-session and between-session agreement and reliability for both types of digital algometers.** Upper parts show agreement and reliability of the Somedic algometer; lower parts show agreement and reliability of the Wagner algometer. Within-session and between-session agreement expressed as coefficient of variation (CV), including 95% Confidence Intervals. A lower percentage in the coefficient of variation indicates a better agreement. Average PPT's are given per location in kilo Pascal. Reliability is expressed as intraclass correlations coefficients, ICC$_{1,1}$, including 95% Confidence Intervals. Dotted horizontal lines represent classification of *Koo & Li (2016)*: values ≥0.75 indicate good reliability and ≥0.9 excellent reliability.

maximal offset was −55 and +39 kPa (both on the tibialis anterior muscle). The CV ranged from 3.6% to 26.5%, and differences ranged from −9.5 to +6.6 percentage points compared to the group findings. For example, on the tibialis anterior muscle using the Wagner algometer, rater 3 had a CV of 3.6% (95% CI [2.4–7.5%], on an average PPT of 595 kPa) and the CV of rater 6 was 17.8% (95% CI [13.3–27.6%], on an average PPT of 645 kPa). Overall, most raters showed acceptable agreement scores, only rater 9 scored a CV of 26.5% (95% CI [18.3–50.8%], on an average PPT of 304 kPa) on the extensor carpi radialis brevis muscle. In general, the extensor carpi radialis brevis muscle showed the most

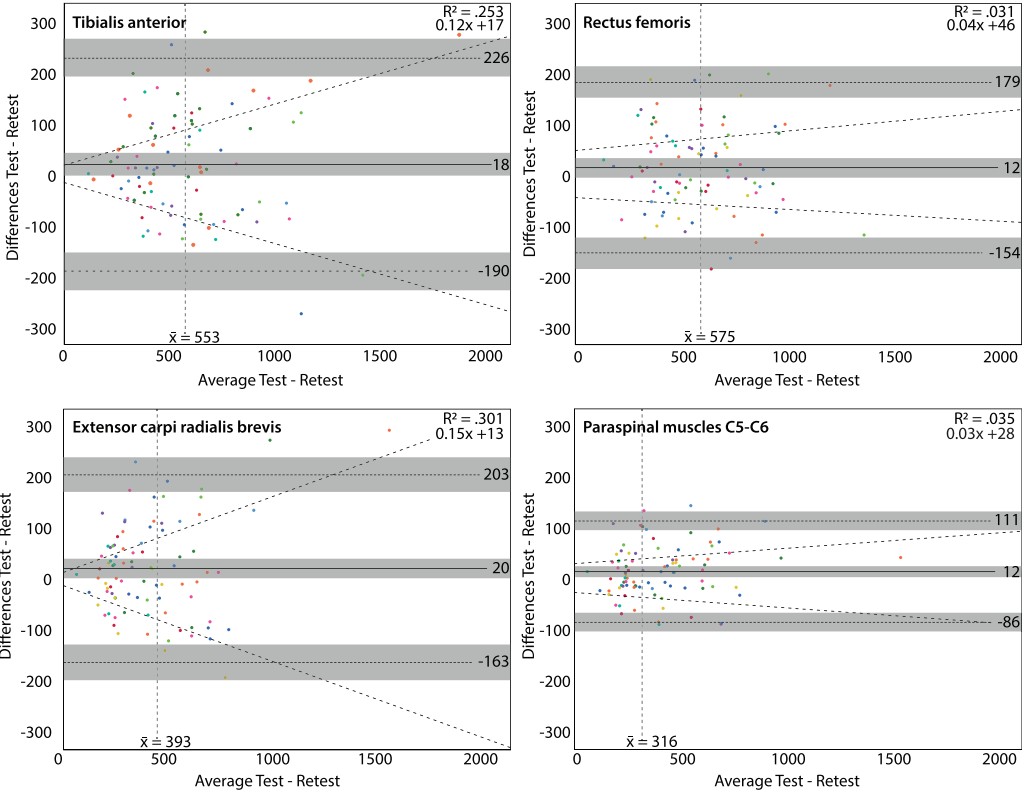

**Figure 3 Bland & Altman plots for the Somedic algometer.** Bland & Altman plots with the average PPT in kilo Pascal of the test and retest on the x-axis and the differences between the test and retest on the y-axis. Grey areas are the 95% confidence intervals of the upper limit, lower limit and systematic bias. The vertical dashed line is the mean threshold for that location. The diagonal dashed line is the linear regression line indicating potential dose-dependent differences. Dose-dependent differences were observed for the tibialis anterior muscle (upper left panel) and extensor carpi radialis brevis muscle (lower left panel).

unfavourable agreement scores, probably due to the smaller size increasing the complexity of the measurement (*e.g.*, more difficult to localise and more difficult to keep the algometer on the muscle during the measurement). All individual rater reproducibility parameters are shown in Fig. 5 and are presented in Appendices A and B.

Between-session agreement and reliability were comparable for males and females except for the tibialis anterior muscle using the Wagner algometer (Appendix C).

## DISCUSSION

This study showed good intra-rater reproducibility parameters for novice raters after 8 h of training, including 6 h of performing PPTs, with a Somedic and Wagner digital algometer (*i.e.*, 3 h per algometer). Evaluation of individual intra-rater, between- and within-session agreement showed small differences per rater, but within an acceptable range.

The reproducibility parameters found for novice raters were comparable or even slightly better than those reported for experienced raters (*Sterling et al., 2002*; *Ylinen et al., 2007*; *Jones, Kilgour & Comtois, 2007*; *Jørgensen et al., 2014*; *Waller et al., 2016*; *Jakorinne, Haanpää & Arokoski, 2018*; *Srimurugan Pratheep, Madeleine & Arendt-Nielsen, 2018*;

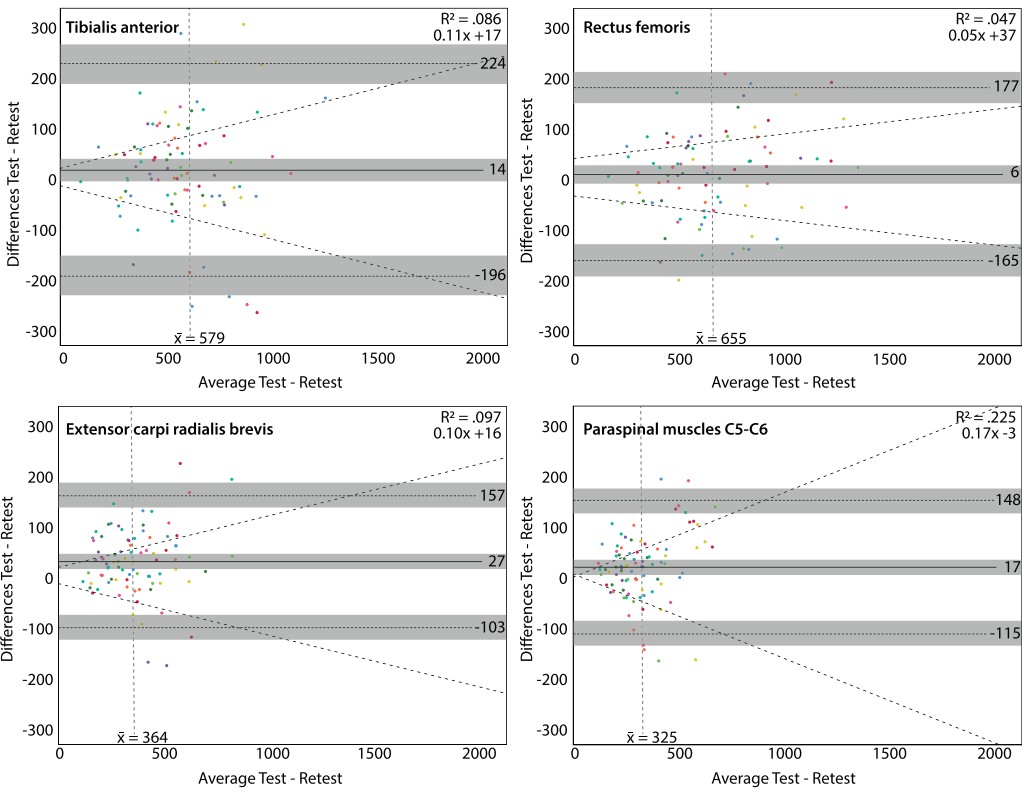

**Figure 4  Bland & Altman plots for the Wagner algometer.** Bland & Altman plots with the average PPT in kilo Pascal of the test and retest on the x-axis and the differences between the test and retest on the y-axis. Grey areas are the 95% confidence intervals of the upper limit, lower limit and systematic bias. The vertical dashed line is the mean threshold for that location. The diagonal dashed line is the linear regression line indicating potential dose-dependent differences. Dose-dependent differences were observed for the paraspinal muscles (C5-C6) (lower right panel).

*Knapstad et al., 2018*; *Middlebrook et al., 2020*). Between-session agreement was comparable, depending on the location. In most locations, this study revealed equal or better CVs (*e.g.*, ~22% at the extensor carpi radialis brevis muscle compared to 16% in our study), except at the tibialis anterior muscle. A previous study found a slightly better CV at the tibialis anterior muscle in experienced raters (8% *vs.* 10–14%) (*Sterling et al., 2002*) compared to our study. In contrast, these authors found slightly lower ICC values for reliability (ICC 0.65 to 0.94) compared to our study (ICC 0.87 to 0.95). The mean PPTs we obtained at the different locations were comparable with previously reported values (*Sterling et al., 2002*; *Ylinen et al., 2007*; *Jones, Kilgour & Comtois, 2007*; *Jørgensen et al., 2014*; *Waller et al., 2016*; *Jakorinne, Haanpää & Arokoski, 2018*; *Srimurugan Pratheep, Madeleine & Arendt-Nielsen, 2018*; *Knapstad et al., 2018*; *Middlebrook et al., 2020*).

One other study included novice undergraduate physiotherapy students, using a Wagner algometer at the tibialis anterior muscle and paraspinal muscles in a comparable population of healthy, young participants (mean of 25.4 years) (*Walton et al., 2011*). They found slightly lower reliability (ICC 0.79 *vs.* 0.88 in our study) and within-session agreement values (CV 19% *vs.* 13% in our study). The lower reliability and within session

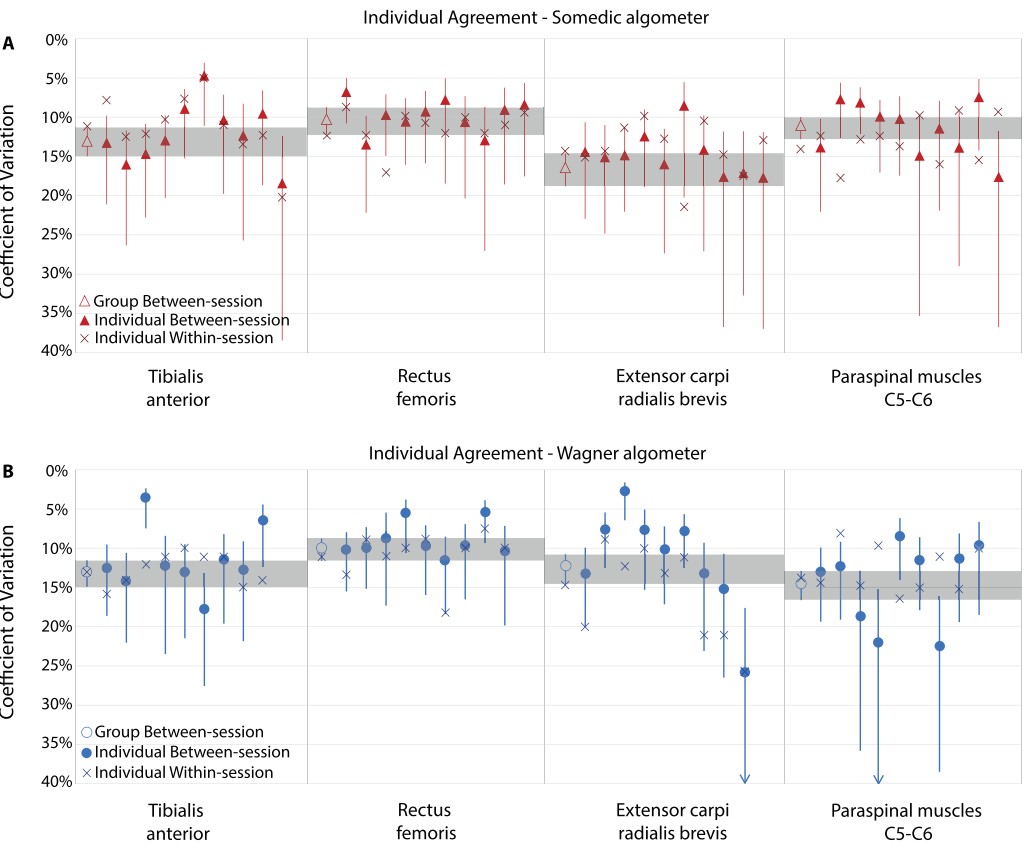

**Figure 5 Individual exploration of reproducibility parameters of novice raters.** Agreement expressed as coefficient of variation (CV), including 95% Confidence Intervals, of the Somedic algometer (A) and Wagner algometer (B). Shaded areas show the overall group between-session agreement confidence interval as reference. Each filled mark represents an individual rater.

agreement that they found could probably be explained by the lower number of training hours compared to our study (1 *vs.* 8 h) (*Walton et al., 2011*). Earlier studies also attributed lower individual reliability scores to systematic errors from novice raters (*Chung, Um & Kim, 1992*; *Goulet, Clark & Flack, 1993*). Due to potential different learning curves (not every rater learns equally fast and the number of participants measured to reach acceptable reproducibility may differ), the training could have a variable effect on each rater's performance (*van Duijn, Swanick & Donald, 2014*). Additionally, it is unknown how much training is needed to perform PPTs with an acceptable reproducibility.

In this study, an arbitrary number of ten raters was chosen, as we hypothesised that individual (psychomotor) skills could influence the reproducibility parameters. Using a group of multiple raters increased the generalisability of the findings and allowed exploration of differences in the reproducibility parameters of individual raters. Even though novice raters show good group reproducibility parameters compared to experienced raters, individual differences in novice were identified. No previous studies assessed individual reproducible parameters in experienced or novice raters. Multiple raters showed a CV and SEM which were over two times smaller than the group's CV and

SEM, as such they have excellent measurement skills techniques compared to their peer raters.

This study has some limitations. Ideally, for exploration of rater differences, all raters should have tested the same group of participants to minimise the subject variability. However, different subsamples were chosen to reduce the number of tests per participant and prevent potentially attention bias, salience and sensitisation effects (*Villemure & Bushnell, 2002*; *Goffaux et al., 2007*; *Hall & Rodríguez, 2017*; *Moore et al., 2019*). Consequently, we could not directly compare reproducibility values between the two different algometers. However, the two samples were comparable based on demographics and PPT values. Another limitation is that not all raters measured an equal number of participants (min = 6; max = 14), which influenced the width of the confidence intervals and the amount of experience gained while acquiring the reproducibility data. Remarkably, raters who measured a higher number of participants showed noticeable higher reproducibility parameters. This could possibly be explained by the differences in gained experience through measuring. The individual parameters should, however, be interpreted with caution due to the low subsample sizes.

We found some individual differences in reproducibility values, which were within acceptable ranges. Consequently, novice raters who attended an 8-h training can participate as raters in research projects and use PPTs in clinical practice adequately. Since individual differences in reproducibility parameters may exist, researchers and clinicians should be cautious when using reproducibility parameters of other raters from former studies. Future studies should focus on quantifying these differences and include methods to explore whether rater experience or psychomotor skills (*e.g.*, strength or dexterity) may explain these differences. Furthermore, the relationship between the duration of training and reproducibility parameters should be explored further to recommend the minimal duration of training needed to perform PPTs adequately.

In conclusion, although slight differences between individual assessors exist, this study revealed that novice raters show good reproducibility parameters in determining PPTs across four body locations following 3 h of practice of PPT measures in addition to surface palpation skills.

### Funding
This study was conducted with a research grant for teachers of the Dutch Organisation of Scientific Research (NWO) under project number 023.011.069. The funders had no role in study design, data collection and analysis, decision to publish, or preparation of the manuscript.

### Grant Disclosures
The following grant information was disclosed by the authors:
Dutch Organisation of Scientific Research (NWO): 023.011.069.

## Competing Interests

The authors declare that they have no competing interests.

## Author Contributions

- Roland R. Reezigt conceived and designed the experiments, performed the experiments, analyzed the data, prepared figures and/or tables, authored or reviewed drafts of the article, and approved the final draft.
- Geranda E. C. Slager conceived and designed the experiments, performed the experiments, authored or reviewed drafts of the article, and approved the final draft.
- Michel W. Coppieters conceived and designed the experiments, analyzed the data, prepared figures and/or tables, authored or reviewed drafts of the article, and approved the final draft.
- Gwendolyne G. M. Scholten-Peeters conceived and designed the experiments, analyzed the data, prepared figures and/or tables, authored or reviewed drafts of the article, and approved the final draft.

## Human Ethics

The following information was supplied relating to ethical approvals (*i.e.*, approving body and any reference numbers):

The study was approved by the Medical Ethical Committee of the University Medical Center in Groningen, The Netherlands (METc 2016.613; M17.207169).

## Data Availability

The SPSS data of PPT measurements is available in the Supplemental Files.

## Supplemental Information

Supplemental information for this article can be found online at http://dx.doi.org/10.7717/peerj.14565#supplemental-information.

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
