# Peer review of "Novice assessors demonstrate good intra-rater agreement and reliability when determining pressure pain thresholds; a cross-sectional study"

_PeerJ, doi:10.7717/peerj.14565_

## Round 0.1 · original submission · Major Revisions

The reviewers raised some critical issues that the authors should carefully consider. We provide a chance to improve the document, but the changes should be done with great depth.

The main weakness of this research is the replicability of the study design (i.e., it is not adequately described in the paper). Moreover, statistical analysis is broad but without any logical pathway, e.g., from differences within and between groups + to absolute and relative reliability. The last important issue is a very high CV, without any explanation.

·

Basic reporting

Original title:
Novice raters demonstrate good intra-rater agreement and reliability when determining pressure pain thresholds; a cross-sectional study
Authors:
Roland R Reezigt, Geranda C. Slager, Michel W. Coppieters, Gwendolyne G.M. Scholten - Peeters

Dear Authors and Editor:
I really appreciate Your work, the topic is really interesting and your paper surely will have significant impact on practice in pain monitoring and assessment. Here below I have some suggestions which may help You increase the quality of work.

The structure and language is very well and professional which gives the reader comfort in going through the whole paper.
The raw data files are supplied and can be opened in SPSS, they are well structured and labeled .
The figures and tables are well presented, (minor suggestions below).

A. Basic reporting

1 Title change:
Consider changing title to:
Novice raters’ demonstrate good intra-rater agreement and reliability when determining pressure pain thresholds; a cross-sectional study.

This should increase the interest to look into and read through the paper to get the most valuable information instead of giving it straight away in the title ;)

2. If this is possible for You it would be very interesting putting in the results of a very experienced rater using this two algometers as a reference to the novice ones.
3. Figure 2 – I know that for the scientists this is well known, but the readers of Your work should also be physiotherapists and trainers in physiotherapy and they they are not always familiar with statistical parameters. What is more – in this particular case the stat.parameters differ in scale interpretation (normal person naturally thinks that higher is better). That’s why I recommend putting in text or under the figure 2 short information – CV – the higher CV means the greater dispersion, ICC – the higher ICC means the greater reliability.
4. Key words – I suggest adding “pressure pain threshold”

5. Spelling, grammar, language check:
- line 123 – delete “was”
-line 140-141 – it is unclear for how much time the raters practiced on different types of algometers – the reader may suppose that f.e. they used wagner for 5,5 h and Somedic for 0,5h – so please specify.
-line 142 – add “main” study.
-line 161 – change to Then, every participant
-line 163 – change to All four rooms were the same in means of temperature, quiet and … .
-line 294 – change to “using both algometers”
-line 328-330 – “Exploration of individual intra-rater, inter-session and within-session agreement some differences per rater, but within acceptable (i.e., -6.0 to +9.3 percentage points)
330 per rater.” This needs reframing because there is no action word in this statement.
Line – 338 – repetition
-line 346 – change “worse” to “lower”
-line 349 – change trainings hours to training hours

6. Please align the text and move single characters like "a" or "in" to the next line of text. Also increase the space between titles of subsection and text.

7. Please check the interpunction through whole manuscript
– text under figure 5 should end with “.” After “rater”.
- Table 1. BMI – Body Mass Index, PPT – Pressure Pain Treshold etc.

Experimental design

B. Experimental design and presentation of results

1. The main limitation of the experimental design is that raters assessed different number of participants – f.e. 6 vs 14. This is like one had more “training” and gaining experience and the other less which could have impact on the results. Of course very strong statistical analysis levels this effect.

2. “Dotted lines represent classification of Koo et al (2016): values g0.75 indicate good reliability
and 0.9 excellent reliability.” This is obvious to me, but… You have different kinds of dotted lines on the different panels of Figure 2. So I suggest using different style of the line for ICC borders and more traditional style for the error bars

3. presentation of the results:
I think that from the general point of view the question what is the general agreement and general reliability is interesting, so I propose putting in the figure 2 a line representing the mean agreement for these 4 examined muscles both for reliability and agreement graphs.
What is more – for a person who is just getting to know the world of algometer testing, the important information is what were the average values of PPTs for a given muscle. With this information, the difference value will be easier to interpret. meanwhile, the article talks about differences all the time, without specifying measured values.

4. table 1
– add unit for alcohol use
- height and weight, BMI – these are not demographics but antrophometrics, sports h/week and alcohol use are neither demographics so please change the structure of the table on demographics
- age
-sex

Antrophometrics or morphological parameters
- height
-weight
-BMI

Quality of life?
-Sports [h/week]
-Alcohol use [unit]

5. table 2
– shouldn’t SEM have a kPa unit?
- change Wagner algometer and Somedic algometer in bold

Validity of the findings

C. Validity of the findings
Strong statistical analysis was conducted using adequate tests and methods which is strongly supported by references from the up-to-date literature. The topic is crucial for scientists and physios using algometers in their research and practice. The further studies directions are clearly stated.

Additional comments

D. General comments.
If You Dear Authors would like to put it in this paper it would be interesting to see whether there are differences between male and female raters. I know that this could be a material for another paper, but even briefly with one graph you can show that there are / there are no differences between male and female raters present in Your study.
I suggest putting in the graphs results of a very experienced rater to be a point of reference to the novice ones.
I suggest putting in the text average results of the PPTs on each muscle to give the reference for the values of differences.

All the best in Your further research!

Reviewer 2 ·

Basic reporting

.

Experimental design

.

Validity of the findings

.

Additional comments

1. Line 97: Why tibialis anterior and neck muscles are referred to? What is the explanation?
2. Line 103: "Determining PPTs in a reproducible manner requires practical and psychomotor skills". This sentence should be re-arranged and improved. Now, it sounds out of context. Please, re-edit this sentence to provide a brief summary of the paragraph above. Additionally, it should consist of a novel perspective, which will be analyzed below.
3. Line 103-113: A clear explanation of the research problem is needed. Moreover, please prepare a clear perspective of the prior studies. And explain, the novelty and importance to establish this study and data.
4. Line 120: The study design should be more precisely described, especially: the model - authors used randomization - described in line 188; describe the reliability model - why not use intra-, inter-rater reliability for absolute and relative value; and intra-algometer reliability for absolute and relative value?
5. Line 129: Please add inclusion/exclusion criteria
6. Line 129: Why 10 raters were recruited? I can not understand the model used for reliability.
7. Line 144: This sub-section should be very detailed and described
8. Line 197: I would suggest moving to the sub-section "Participants"
9. Line 205: Or statistical analysis? I would suggest changing the title of this sub-section
10. Line 210-212: All information about distribution should be included in this part. Furthermore, no information about the statistical criteria for independent analysis is attached
11. Line 210: Please add also information, that comedic was in kPa
12. I am a bit confused about the number of analyses. Would be easier and more clear to add: ICC, SEM, MDC, and LOA? Additionally, to establish differences within- and between groups, an ANOVA could be used.
13. This entire section is not clear to me. I would suggest using a model from #12

---

## Round 0.2 · Minor Revisions

Please consider the statistical comments provided by the second reviewer.

·

Basic reporting

Dear Authors, I appreciate a lot your work to improve the manuscript. All my comments and questions were substantively answered and reflected in the form of changes and improvements in the article. Further, improvement in english, style and technical form of the manuscript are visible.

Experimental design

Despite the fact that the methodology used in the study is subject to discussion, the authors effectively defend their position by answering in a substantive manner and supporting their answers with references to the literature or making required changes.

Validity of the findings

In my view the novelty of this paper is high and brings practical aplication to practicioners and scientists.

Additional comments

I leave the assessment of the statistical analysis to other reviewers and editors who surely have more knowledge and experience in this aspect.

Reviewer 2 ·

Basic reporting

I have marked my concerns in 4. Additional comments.

Experimental design

I have marked my concerns in 4. Additional comments.

Validity of the findings

I have marked my concerns in 4. Additional comments.

Additional comments

1. Introduction - Please connect the last two paragraphs to obtain a clear research perspective.
2. Line 335 - Please identify the exact model and type of ICC according to Koo and Li (2016).
3. Reliability - I am confused about the level of the CV, could you explain this part?
4. Why have you not included MDC90? In my opinion, it would be more appreciated.
Relative reliability (Intra Class Correlation coefficients-ICC), absolute reliability (SEM), MDC,
5. Line 519 - Please discuss the level of CV in your study, and explain these differences compared to previous papers.
6. Line 463 - Please define a clear conclusion.

---

## Round 0.3 · accepted · Accept

The article was improved and the two reviewers agreed in to accept the current version.

·

Basic reporting

I have revised the whole paper again. Looking at the track-changes version number of changes can be seen. The statistical analysis is made more clear at this time.
The only thing I find confusing is the use of words raters-assessors. In the title there is word "assessors". In the abstract section the authors changed "raters" to "assessors" , but in the following sections the word raters stil remain (line 164 - f.e. - "raters" is the title of the subsection). So, please decide and be consistent in this question

Experimental design

After previous reviews and making changes by the authors I have no more remarks.

Validity of the findings

After previous reviews and making changes by the authors I have no more remarks.

Additional comments

After previous reviews and making changes by the authors I have no more remarks.

Reviewer 2 ·

Basic reporting

The authors improved their manuscript.

Experimental design

The authors improved their manuscript.

Validity of the findings

The authors improved their manuscript.